# Micro/Nanorobots for Medical Diagnosis and Disease Treatment

**DOI:** 10.3390/mi13050648

**Published:** 2022-04-19

**Authors:** Yinglei Zhang, Yuepeng Zhang, Yaqian Han, Xue Gong

**Affiliations:** 1College of Light Industry, Harbin University of Commerce, Harbin 150028, China; kahnannie@163.com; 2Clinical Medical College, Harbin Medical University, Harbin 150081, China; zhangyuepeng93@hrbmu.edu.cn; 3School of Instumentation Science and Engineering, Harbin Institute of Technology, Harbin 150001, China; yqhan1995@126.com

**Keywords:** micro/nanorobots, driving mechanism, medical diagnosis, disease treatment

## Abstract

Micro/nanorobots are functional devices in microns, at nanoscale, which enable efficient propulsion through chemical reactions or external physical field, including ultrasonic, optical, magnetic, and other external fields, as well as microorganisms. Compared with traditional robots, micro/nanorobots can perform various tasks on the micro/nanoscale, which has the advantages of high precision, strong flexibility, and wide adaptability. In addition, such robots can also perform tasks in a cluster manner. The design and development of micro/nanorobots and the integration of surface functionalization, remote drive system, and imaging tracking technology will become a key step for their medical applications in organisms. Thus, micro/nanorobots are expected to achieve more efficient and accurate local diagnosis and treatment, and they have broad application prospects in the biomedical field. This paper aims to introduce relevant driving methods of micro/nanorobots preparation in detail, summarizes the progress of research in medical applications, and discusses the challenges it faces in clinical applications and the future direction of development.

## 1. Introduction

The emergence of micro/nanorobots promotes the development of a precision medicine, which is an important direction of modern biomedical development. Micro/nanorobots refers to the functional devices that can realize motion at micron and nanoscale, driven by a light field, magnetic field, and sound field [1]. Richard Phillips Feynman first suggested that micro/nanorobots can be used in biomedical applications, and he predicted that the micro-machines can achieve microscopic treatment [2]. Professor Toshio Fukuda, the father of micro/nanorobots in the early 2000s, pioneered the world’s first nanorobot operating system for single-cell analysis and manipulation [3]. However, due to the difficulty of manufacturing micro/nanomaterials with complex structures of different physical properties, the development of micro/nanorobots faces great challenges. In recent years, through top-down and bottom-up methods, the artificially synthesized micro/nanorobots have not only achieved a breakthrough from centimeter level to micro/nano level but also developed micro/nanorobots with various materials and structures, such as tubular, linear, rod, yin-yang spherical, spiral, peanut, and sea urchin micro/nanorobots. Due to its small structure and controlled navigation ability, micro/nanorobots have been widely used in many fields, including drug targeting delivery, cell capture and separation, minimally invasive surgery, analysis and detection, environmental purification, and nano printing [4,5]. With the deepening of research, the motion control methods for micro/nanorobots are also developing. For example, the navigation motion of micro/nanorobots can be realized by using an electric field, magnetic field, ultrasound, and light field.

With the continuous development of micro-nanomaterial synthesis technology, various micro/nanorobots have been developed in depth and have broad application prospects, especially in the biomedical field [6,7]. Biomedicine mainly includes diagnosis and treatment. From an engineering point of view, diagnosis is the measurement of various abnormal phenomena in the human body, and treatment is to change the existing state of human cells. With the development of medicine to the cellular and molecular stages, diagnostic techniques need to be reduced to micron and nanoscale scales, sensing at the cellular molecular scale, and providing new diagnostic methods [8]. The same is true of the treatment, which begins at the cellular level and at the DNA molecular level, whereas DNA molecules are the core code for controlling cell growth and development, and cells are the basic units that make up organ tissue, which will solve human disease problems for more fundamental reasons [9]. Therefore, the smaller scales of sensing and manipulation methods are required in diagnosis and treatment. The emergence of robots is the innovation of modern biomedical methods, providing new ideas, which can enter the human body in minimally invasive ways, and none of the traditional medical technology is able to achieve this technology [10]. Although medical micro/nanobots have made some progress over the past decade, immature technology requirements in this field, such as the drive and cluster control of microrobots, limited the widespread clinical use of these tools, while driving a certain number of micro/nanobots often requires cumbersome procedures and advanced instruments [11].

This paper briefly highlights the dynamic foundation of micro/nanorobots, summarizes the latest trends in micro/nanorobots research, and focuses on their applications in medical diagnosis and disease treatments, see Figure 1, and the major challenge of medical micro/nanorobots is from the laboratory to clinical application.

## 2. Preparation Method of Micro/Nanorobots

The manufacturing methods of micro/nanorobots are mainly divided into top-down and bottom-up. Among them, top-down strategy includes physical vapor deposition (direct deposition and grazing angle deposition), roll-up technique, and laser direct writing 3D printing technology. The strategy of bottom-up includes template electrochemical deposition technology and wet chemical synthesis technology, which will be introduced one-by-one in the following.

### 2.1. Physical Vapor Deposition Technology

Physical vapor deposition needs to be carried out in an ultra-clean vacuum environment. The main process is that the massive target is transformed into gaseous atoms and molecules, which are deposited on the sample surface to form nano films. Common physical vapor deposition methods include vacuum thermal evaporation, magnetron sputtering, and electron beam evaporation. These technologies can not only deposit common metal and alloy films but also realize the preparation of functional material films, such as semiconductors and ceramics. Physical vapor deposition is mainly divided into direct deposition and grazing angle deposition. Among them, direct deposition is usually used to process the Janus microsphere robot, i.e., the microsphere is tiled on the substrate surface, and the functional film material is directly deposited on the upper surface of the microsphere, while the functional film cannot be deposited on the lower half of the microsphere because it is blocked, so the Janus microsphere robot can be formed. As shown in the figure, Chen et al. [12] prepared the microsphere robot of titanium dioxide platinum bifunctional film in batch and realized the controllable “start and stop” of the robot based on the chemical/optical synergy. The grazing angle deposition technique is a physical deposition method with dynamic tilt angle. Due to the self-shadowing effect, gaseous atoms suppress their own surface diffusion, and complex nanostructures can be prepared with real-time switching of deposition angles.

### 2.2. Self-Winding Technology

The self-coiling technology is to deposit multi-layers of different materials, in turn, and, in this process, different stresses are preset by using the strain difference of the material itself, and then the asymmetric stress is released after the sacrificial layer is etched by wet etching. Finally, under the action of stress, the self-coiling structure is spontaneously coiled into a tubular or spiral structure. Mei et al. [13] used self-coiling technology to realize the coiling of various inorganic materials (Pd/Fe/Pd, TiO_2_, ZnO, Al_2_O_3_, SiN/Ag, and C) films and, thus, prepared a variety of tubular micro/nanorobots with different material components. In addition, Zhang et al. [14] processed InGaAs/GaAs/Cr composite micro-screw robots by multi-step photolithography, reactive ion etching (RIE), and wet etching to release prestress.

### 2.3. Laser Direct Writing 3D Printing Technology

Laser direct writing 3D printing technology is based on the optical polymerization effect to process any 3D microstructure with high resolution, and then the micro/nanorobots can be effectively driven by physical vapor deposition or adsorption of functional materials (Ni, Fe, and Fe_3_O_4_). Li et al. [15] realized the preparation of fishtail-like micro/nanorobots based on femtosecond laser polymerization, as well as realized the self-driving of bubbles in hydrogen peroxide by physical vapor deposition of Pt. It is worth noting that the multi-channel of the fishtail allows bubbles to jet in parallel, which greatly improves the speed and thrust of the microrobot.

### 2.4. Electrochemical Template Deposition Technology

Electrochemical deposition is a process that metal or alloy is deposited from the corresponding salt solution by redox reaction. Electrochemical deposition has the advantages of simple operation and low cost, which is very suitable for the preparation of large quantities of micro/nanorobots. By selecting different auxiliary templates, various micro/nanorobots with different shapes (tubular, rod, and spiral) can be deposited. Among them, porous alumina membrane (AAO) is the most common auxiliary template for electrochemical deposition. Specifically, before electrochemical deposition, a metal electrode film was sputtered on the surface of AAO, and then it was put into the electrolyte for the target deposition of metal. After redox reaction, the deposited micro/nano structures were obtained. Then, the deposited AAO was put into nitric acid and sodium hydroxide solutions to remove the metal electrodes and templates, respectively. Finally, the prepared micro/nanorobots were collected. Gao et al. [16] used porous aluminum film to deposit hollow cone-shaped micro/nanorobots, achieved/self-driven by decomposing oxygen peroxide jet bubbles through metal Pt in the cone-shaped cavity.

### 2.5. Wet Chemical Synthesis 

The main process of wet chemical synthesis is to reduce the compound solution of the target material to simple structures, such as spherical, rod-like, and peanut-like, by chemical reactions, and the controllability of the structure obtained by this method is relatively poor. Lin et al. used the hydrothermal synthesis method to mix FeCl_3_, NaOH, and Na_2_SO_4_ in an electric oven at 100 °C for seven days to prepare a magnetic peanut-like micro/nanorobots. Driven by an external magnetic field, the micro/nanorobots realized two motion modes of rolling and rolling. In addition, Dai et al. [17] prepared Janus (Si, TiO_2_) rod-like self-electrophoresis micro/nanorobots by wet etching and hydrothermal synthesis, which showed good phototaxis after modification.

## 3. Driving Mode of Micro/Nanorobots

Micro/nanorobots can flow both individually and in clusters. The driving mode will affect micro/nanorobots locomotion speed, controllability, and biocompatibility, thus affecting application in biological organisms. To overcome the challenges in low Reynolds fluid, the active motion of micro/nanorobots mainly relies on the conversion of converting a local chemical (e.g., H_2_O_2_, urea, etc.), or physical energy (e.g., light, ultrasound or magnetic fields, etc.), and by microorganisms or cells (e.g., sperm, etc.) toward mechanic propulsion [18].

### 3.1. Chemical Propulsion

In nature, the protein biomotors, such as mysin, exert the intracellular propulsion though catalytic decomposition of adenosine triphosphate (ATP). Inspired by them, a variety of micro/nanorobots, consisting of catalysts, have demonstrated the high propulsion with chemical reactions during the past decade. Among them, the role of catalyst is to react with fuel on the surface of the robot, while inert material is used to construct asymmetric structure. H_2_O_2_ is the first and most widely studied fuel. Micro/nanorobots can generate self-electrophoresis mechanisms and drive or use materials, such as their own platinum (Pt), to catalyze the decomposition of H_2_O_2_ to create bubbles, promoting their own movement (Figure 2a) [19,20]. High concentrations of H_2_O_2_ are strong and incompatible with organisms. Therefore, in order to realize practical applications, especially when chemical driving is used to drive micro/nanorobots in biological systems, it is necessary to develop new in-situ fuels other than H_2_O_2_, i.e., the raw materials should be natural substances in biological fluids. For instance, using biodegradable Zn or Mg, hydrogen can be generated by reacting with the acidic environment of the stomach to achieve self-propulsion and leaving a non-toxic product [21], and a catalytic reaction by replacing Pt with an enzyme, so that the fuel can be replaced with a variety of biomolecules, such as glucose or urea [22].

### 3.2. Physical Propulsion

Most of the external field-driven micro/nanorobots do not require fuel and are driven primarily by light, ultrasonic, or magnetic fields, making them both biocompatible and sustainable, and they are more flexible in terms of control movement. It is relatively easy to establish acoustic conditions. Sonic waves can propagate through solid, liquid, and air media, i.e., deep into the biological tissue, triggering the micro/nanorobots from the outside, without damaging the human body. The light-driven micro/nanorobot is constructed of photoreactive materials, mainly including photocatalytic materials, photorated materials, and photothermal materials. With the irradiation of light, these photoactive materials can absorb light energy to initiate photocatalytic reactions, photopolymerization reactions, and photothermal conversion reactions [23]. The optical drive method is simple to operate and responsive. As shown in Figure 2b, since ultraviolet-visible light is limited to the penetration depth of the semiconductor particles, the photo-driven isotropic TiO_2_ molecular robot can generate non-electrolyte O_2_ molecules by photocatalytic decomposition of H_2_O_2_. This penetrating O_2_ gradient can generate oriented local gradients, driving the TiO_2_ microcircuit to move to the low O_2_ concentration region. The direction of motion is always in the direction of irradiation light, similar to the tendency of microorganisms [24]. In addition to providing driving force, near-infrared light has the potential for optical imaging to track the movement of micro/nanobots in the body [25]. Ultrasonic driving a mechanism of a nano-shaped robot is under the action of ultrasonic waves, and local inactive stream stress on the surface of the asymmetric nanorobot produces a driving force for motion [26]. A bionic nanorobot consisting of the red blood cell membrane and platelet membranes with gold nanowires, showing fast and effective long acoustic advancement in whole blood, can imitate natural activity cell movement (Figure 2c). This mechanism enhances the combination of micro/nanorobots on pathogen and toxins and improves toxicity efficiency [27]. In addition, high-intensity focus ultrasonic waves can also be used to induce rapid evaporation of chemical fuel and generate tubular micro/nanorobots in bullet motion. This microtube can move with a high average speed and penetrates the tissue with a powerful thrust [28]. Magnetic field-driven micro/nanorobots can also be driven without any fuel addition without harm to the human body. The driving condition of magnetic field is usually to use magnetic materials to construct micro/nanorobots to respond to an external magnetic field. Among them, the external magnetic field is divided into a rotating magnetic field, gradient magnetic field, and oscillating magnetic field. For example, the remote controllability and precise motion of three-dimensional magnetic tubular robots, which have good capabilities in the capture, targeted delivery, and release of SiO_2_ particles, can be achieved through external magnetic fields [29]. Except that, the sports metal-organic framework (MOF) is a potential candidate for small- and medium-sized machine-man platforms for environmental remediation, targeted drug delivery, and nanosurgery (Figure 2d). Spiral micro/nanorobots with biocompatibility and pH response characteristics prepared by Wang et al. are coated by Zn-based MOF and zeolite imidazole frame 8 (ZIF-8) [30]. These highly integrated multi-function micro/nanorobots can move along a pre-designed track under the control of weak rotation magnetic fields and achieve efficient delivery of goods in the complex microfluidic channel network. In addition, the magnetic field can also control the movement of micro/nanorobots, together with other physical fields. Xu et al. [31] proposed a reliable propulsion method to reduce excess lateral drift during the entire motion by applying ultrasonic waves to the substrate, which is a novel and effective strategy to further improve motion control, as long as an electromagnetic coil system is established to drive the spiral robot.

**Figure 2 micromachines-13-00648-f002:**
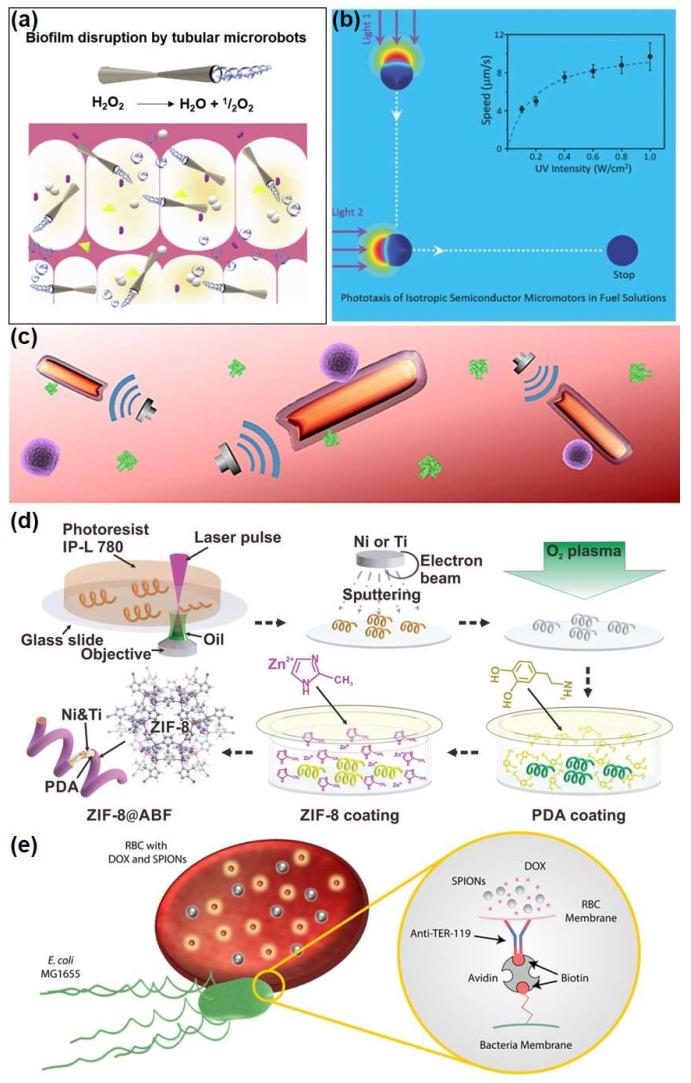
Different driving mode of micro/nanorobots. (**a**) Self-propelled microbots with high antibacterial activity for the degradation of dental biofilm [19], copyright 2020, Elsevier; (**b**) a disruptive strategy to design micromotors by using isotropic structures [24], copyright 2016, Wiley-VCH; (**c**) schematic of biomimetic robots for binding and removal of threatening biological agents [27], copyright 2018, American Association for the Advancement of Science; (**d**) schematic illustration of the component steps involved in the manufacture of ZIF-8@ABF microrobots [30], copyright 2019, Wiley-VCH; (**e**) RBC microswimmers for active cargo delivery, copyright 2018, Wiley-VCH.

### 3.3. Biological Propulsion

Bio-driven micro/nanorobots mainly refer to bio-mixed microbots, which are made up of active microorganisms (cells) and human materials. Microbes, such as bacteria and sperm, which drive their own movement through whiplash, can be used as engines to propel bio-mixed microbots, with sperm also having a unique ability to fuse with somatic cells, which significantly improves the biocompatibility and safety of micro/nanorobots [32]. For example, a bio-mixed microrobot system with motion sperm cells as a power source and drug carrier includes a 3D-printed magnetic tubular microstructure with four arms. Compared to pure synthetic micro/nanorobots or other vectors, this sperm hybrid micro/nanorobot can encapsulate high concentrations of drugs in the sperm membrane, thereby protecting them from the effects of body fluid dilution and enzyme degradation [33]. In addition, *Escherichia coli* (*E. coli*) is also a commonly used biological driving source. Alapan et al. [34] used *E. coli* as a power source combined with magnetically coded red blood cells, demonstrated in Figure 2e, under the control of the magnetic field using its controlled motion to complete the loading and targeted release of drugs.

In short, each drive mode has its own unique advantages. Chemical- or bio-driven micro/nanorobots utilize built-in energy conversion, or autonomous energy, of microorganisms, while physical-driven micro/nanorobots move as a result of interactions between the external field, the structure of the robot, and the medium in which it operates.

## 4. Micro/Nanorobots in Medical Diagnosis

Recent studies have shown that micro/nanorobots have powerful functions in the field of purification, detoxification, and sensing detection. The controllable driving characteristics enable microrobots to actively search for target objects to be cleared, which greatly improves the detection and removal efficiency.

### 4.1. Sensing Detection

Micro/nanorobots can mix with fluids and induce target receptors to interact, providing the possibility for medical diagnosis. Micro/nanorobots can selectively identify metal ions, bacterial toxins, proteins, cells, etc., offering accurate pre-treatment analysis for disease treatment. Accurate detection of metal ions in the blood can prevent excessive ion concentration to protect human health [7,35,36]. For example, a new type of magnetic mesoporous silica/ZnS·Mn/gold/tetraethylenepentamine/heparin (MMS/ZM/Au/T/Hep) micromotors can directly detect and remove excess copper from the blood. The micro/nanorobot accelerated the spread of solute and fully mixed with the target, and its magnetic meconium silica microtube provided a rich load space for the adsorption of tetramethylene pentaamine (TEPA), thus showing good adsorption ability and short processing time for Cu^2+^. Due to the synergy of mesoporous structure, adsorption function group, and good mobility, the removal rate of blood copper ions was as high as 74.1%. At the same time, the micro/nanorobot selectively monitored the concentration of copper ions in the blood based on changes in fluorescent signals after separation from the blood. The entire self-mixing process can be achieved by autonomous motion of the micro/nanorobot without stirring or ultrasound. The magnetic Fe_3_O_4_ enables the micro/nanorobot to quickly separate after removing Cu^2+^ from the blood. This research result provided some support for the integration of toxin detection and removal in the blood, as well as solved the problem of long treatment cycle, high cost, separation of diagnosis and treatment, and limited therapeutic effect of traditional treatment methods. A diagram of the synthetic micro/nanorobot and the removal/detection of Cu^2+^ in the blood is shown in Figure 3a [35].

To diagnose sepsis early on, Molinero-Fernández et al. developed a fluorescent immunoassay based on micro/nanorobot and used it for the determination of calcitonin-lowering protones (PCT). The polypyrrole (PPy) layer of this micro/nanorobot has high binding specific antibodies that actively identify PCT antigens through magnetic guidance and catalyzing the push of bubbles. Within the clinically relevant concentration range, the assay can be used with a small number of samples to measure PCT levels in clinical samples of low-weight newborns suspected of sepsis [36].

In addition, micro/nanorobots, serving as biosensors, have great potential in intelligent sensing and drive systems. Kong et al. [37] have introduced an Mg/Pt Janus miniature robot capable of self-renewing the surface. With the aid of the Mg/Pt Janus micro/nanorobot, the electrochemical detection of glucose in human serum can be improved without the need for additional toxic fuels or surfactants. The study showed that the rapid movement of the Mg/Pt Janus micro/nanorobot enhanced the current signal, as well as the current signal increased with the increase of introduced micro/nanorobots, and the addition of micro/nanorobots established great linear relationship between the current signal and the glucose concentration. Compared to synthetic sensors, red blood cell biosensors and micro/nanorobots are highly biocompatible, flexible, and noninvasive in biological systems. Li et al. [38] used in vivo red blood cell waveguides to build living biosensors and miniature robots, which were limited by optical gradient forces to the optical axis of two conical fibers. Red blood cell waveguides can rotate continuously as microbots to transport particles in the blood in a controlled manner, as shown in Figure 3b. The red blood cell waveguide has been successfully assembled and worked in zebrafish blood vessels, its light transmission mode is sensitive to the surrounding environment and related to the erythropoietin morphology, and the red blood cell morphology depends on the pH of the blood, so the red blood cell waveguide can be used for the body’s acidity sensing, detection of blood diseases caused by acidity and alkalinity, and measurable pH of 5.0 to 9.0. For the sensing of the organism pH, Zheng et al. were inspired by the process of starfish preying on shellfish and constructed a starfish-like microrobot with this non-horomorphized hydrogel, which enables its flexible tentacles to be captured and released in liquid environments with autonomic deformations that effectively fit the outer contours of any target. Through non-uniform electric field cross-link curing, a hydrogel network with different densities is embedded in the single-layer film structure of a brown alginate microrobot, which can sense pH environment (gastric pH: 1–3, intestinal pH: 6–8) automorphosis after the microrobot enters a living digestive system, such as in the human body [39].

### 4.2. Imaging of Medical Micro/Nanorobots

A key clinical application of medical micro/nanorobots is to rely on individuals or groups for monitoring. They can be easily located and guided in the body, and even send signals to induce trigger release, so the potential for medical imaging cannot be ignored. For example, the micro/nanorobot is well detected in the in vivo environment at a penetration depth of 2 mm, and its real-time position in a mouse vein is monitored using optical coherence tomography imaging to provide feedback on the movement of the micro/nanorobot in vivo [40]. Another way is to design a living intestinal micro/nanorobots guided by a photoacoustic computed tomography (PACT) as an imaging contrast agent and a controlled drug carrier (Figure 3c). Among them, the micro/nanorobot has a multi-layer functional coating, and its Au layer enhanced the optical absorption and improved the propulsion rate, gelatin hydrogel layer expanded the load capacity of different functional components, and polyethylene layer maintained the robot’s geometry in the process of propulsion. Because of its high space-time resolution, noninvasiveness, high molecular contrast, and strong depth penetration, the migration of micro/nanorobots capsules to the target area can be observed in real-time [41]. In addition, Iacovacci et al. [42] have proposed a magnetically-driven therapeutic microbot made of thermally responding double-layer hydrogels. The tiny robot contains magnetic nanoparticles and radioactive compounds that act as imaging agents in a hydrogel frame. Magnetic nanoparticles can be used to remotely drive and trigger shape changes in micro-devices, while imaging agents can monitor micro/nanorobots in the body. The epithelial imaging of the mouse was followed by a single-photon emission tomography scan, and the micro/nanorobots were detected in the mouse’s abdomen. For the first time, the study showed that imaging of a single micro/nanorobot can be performed using hydrogel structures as low as 100 μm in diameter, laying the foundation for the future development of single-robot closed-loop control. In addition to photoacoustic imaging, micro/nanorobots can also use magnetic resonance imaging. As shown in Figure 3d, through a simple immersion process, the spiral microcones robot prepared with spirulina microalgae in magnetite suspension has super-smooth magnetism. Since microalgae have the advantages of in vivo fluorescence imaging, the spiral robot showed intrinsic fluorescence, magnetic resonance signals, natural degradation, and ideal cytotoxicity without any surface modification. It can stably navigate in various biological fluids and track noninvasively in superficial tissues or deep organs through autologous fluorescence and magnetic resonance imaging. After subcutaneous and abdominal injections in mice, a micro/nanorobot group was observed in the mice’s stomachs through magnetic resonance imaging [43].

**Figure 3 micromachines-13-00648-f003:**
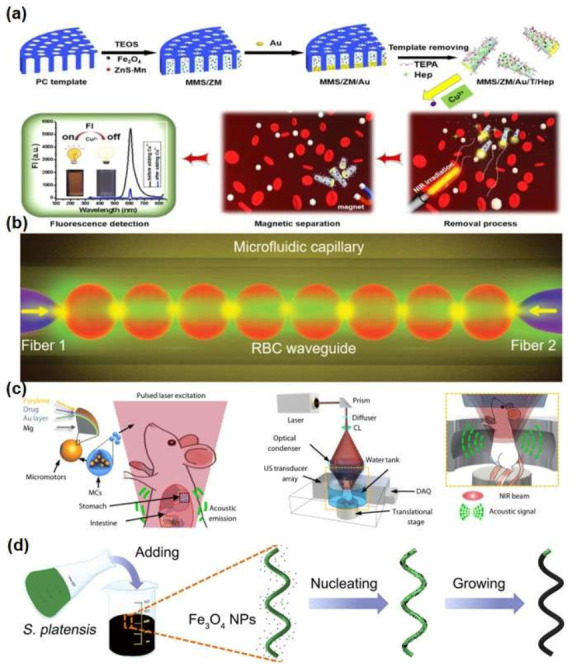
Design and fabrication of micro/nanorobots in medical diagnosis. (**a**) Schematic illustration of the synthesis of MMS/ZM/Au/T/Hep micromotors for blood Cu^2+^ removal/detection treatment [35], copyright 2020, Elsevier; (**b**) schematic illustration showing the optical assembly of the RBC waveguide using two fiber probes in a microfluidic capillary [38], copyright 2019, Wiley-VCH; (**c**) schematic of the PAMR in the GI tract [41], copyright 2019, American Association for the Advancement of Science; (**d**) schematic of the dip-coating process of S. platensis in a suspension of Fe_3_O_4_ NPs [43], copyright 2017, American Association for the Advancement of Science.

## 5. Micro/Nanorobots in Disease Treatment

In recent decades, drug therapy based on nanoparticles has been widely studied. However, drug transport based on nanoparticles usually depends on the circulatory system, lacking the ability of location targeting and barrier permeability to achieve drug delivery in local areas. Compared with nanoparticles, the controllable movement of micro/nanorobots can complete the drug delivery from single cells to local tissues/organs at a cross-scale local area. To sum up, as a new vehicle, the micro/nanorobot has powerful propulsion, accurate guidance, and controllable loading and release characteristics of goods, so that it has broad application prospects in targeted therapy.

In order to improve the efficiency and success rate of surgical operation, the robot system (Da Vinci surgical robot) has been widely used to assist doctors in surgical operation. However, the above methods will still bring huge trauma and long-term recovery to patients. In order to minimize the pain and recovery of patients during surgery, minimally invasive and noninvasive treatment is very important. Thanks to its small size, micro/nanorobots show great advantages in noninvasive treatment. In addition, due to the strong enough propulsion generated by magnetic drive, a micro/nanorobot can penetrate biological tissues for operation, so they can perform local surgical treatment.

### 5.1. Drug Carriers

Therapeutic effects of drugs are usually affected by various factors. Under traditional treatments, the general approach may repeat the drug in high doses if the desired therapeutic effect is achieved, but this is likely to increase toxicity and side effects. The precise delivery potential of micro/nanorobots in the target area is expected to solve the toxicity problem caused by excessive drug use [44]. A drug-loaded micro/nanorobot for the treatment of gastric bacterial infection is made of Mg particles with an average size of about 20 μm, shown in Figure 4a. It has the ability of efficient propulsion, and the average velocity tested in vitro simulated gastric juice (pH = 1.3) is about 120 μm/s. In vitro bactericidal activity test of Helicobacter pylori (H. pylori) showed that the drug-loaded robot demonstrated similar bactericidal activity to free drug solution, and the micro/nanorobot could be effectively promoted and distributed throughout the stomach of live mice, thereby significantly reducing the number of H. pylori. In vivo toxicity studies have demonstrated the safety of micro/nanorobots in the treatment of mouse models. Propulsion of the carrier Mg-microbot in the gastric medium allows for more efficient delivery of antibiotics than passive drug carriers. In addition, the acid-Mg reaction required for autonomous propulsion also consumes protons in the stomach fluid, neutralizing the pH of the stomach [45]. Similarly, another Zn-based micro/nanorobot in gastric drug delivery also has high power propulsion, high load capacity, payload self-release, and non-toxic self-degradation, compared with oral administration of ordinary passive diffusion and dispersion, and its payload retention in the stomach wall has been significantly improved [21]. In addition, the metal-organic framework (MOF) controls drug release through pH response, enabling magnetic movement in cell cultures, drug delivery, and degradation of all its components. Further, a new catalytic micro/nanorobot based on the porous zeolite meth salt skeleton-67 (ZIF-67) is synthetically prepared at room temperature by ultrasonic-assisted wet chemistry [46]. These porous microbots show effective autonomous motion and a long-lasting motion life of up to 90 min in H_2_O_2_. When combined with doxorubicin (DOX), the load can be up to 682 μg/mg. The micro/nanorobot shows excellent drug delivery performance under an external magnetic field due to its porous nature, high surface area, and dual stimulation based on the catalytic reaction of H_2_O_2_ and the effect of H_2_O solvent. Compared with the traditional porous membrane carrier based on the pH response release mechanism, the drug release of the double stimulation-induced porous ZIF-67 microbot is more direct and timelier.

Magnetic fields can precisely control magnetic micro/nanorobots, but the harmfulness of magnetic materials, such as Ni, limits their use in drug delivery. Based on this, Park et al. have developed a biodegradable thermotherapy microbot with a 3D spiral structure and used it to actively control drug delivery, release, and thermotherapy [47]. The microbot is made of poly (ethylene glycol) diacrylate (PEGDA) and pentaerythritol triacrylate (PETA), containing magnetic Fe_3_O_4_ nanoparticles and the cancer drug 5-fluorouracil (5-FU). Under the remote precise control of the rotating magnetic field generated by the electromagnetic drive system, the 5-FU can be released from the micro/nanorobot. Further research on this type of robot found that it responds more to sound energy and releases the drug under ultrasound, as shown in Figure 4b. By changing the condition of the ultrasound beam, it was found that the robot released three modes, natural release, explosive mode release, and constant release of the drug, and the in vitro test results showed that each release mode had different therapeutic results. Among them, in the outbreak and constant release mode, the survival of cancer cells was significantly reduced, confirming that the ultrasound can enhance the treatment effect by increasing drug concentrations and acoustic holes. Ultrasound-mediated therapy can reduce the side effects of the drug because the microrobot can be precisely manipulated to the target position, and the loaded drug can be selectively released by ultrasound focus. Even if some drug transport proteins are offset during operation, drug losses can be minimized by using focused ultrasound active release drugs only at the target location [48]. Based on the excellent magnetism of Fe3O4, Zhong et al. [49] used a hybrid microbiological nanofiltration robot, using microalgae as living scaffolds, to “wear” magnetic coating coat, and target them to tumor tissue, which successfully improved the hypoxic microenvironment of tumors and effectively realized the diagnosis and treatment of tumor under the guidance of three modes of medical imaging: magnetic resonance, fluorescence, and photoacoustic. The photosynthetic biological hybrid micro/nano-swimming body system (PBNs) is to evenly coat superparamagnetic Fe_3_O_4_ nanoparticles to the surface of photosynthetic microalgae Spirulina platensis through the dip coating process to obtain biological hybrid magnetized micro nano-swimming bodies. This hybrid system maintains the efficient oxygen production activity of microalgae and the directional magnetic drive ability of Fe_3_O_4_ nanoparticles. Magnetic engineered PBNs can target and accumulate to the tumor under the control of external magnetic field, and produce oxygen in situ through photosynthesis to reduce the degree of hypoxia in the tumor, so as to improve the efficiency of radiotherapy. At the same time, the chlorophyll released by PBNs after radiation treatment can be used as a photosensitizer to produce cytotoxic reactive oxygen species under laser irradiation to realize collaborative photodynamic therapy. In the mouse orthotopic breast cancer model, enhanced combination therapy showed significant tumor growth inhibition. In addition, PBNs have not only excellent T_2_ mode magnetic resonance imaging function brought by Fe_3_O_4_ coating but also chlorophyll-based natural fluorescence and photoacoustic imaging functions, which can noninvasively monitor tumor treatment and changes in tumor microenvironment. More importantly, the micro nano-swimming body can be effectively degraded in vivo, which provides a transformation prospect for the application of biological hybrid materials in targeted delivery and biomedicine in vivo. Micro/nanorobots can also transport biological agents (e.g., viral vaccines) to treat metastatic tumors in the abdominal cavity (e.g., ovarian cancer). In vitro cell studies have shown that micro/nano-machines can prolong the interaction time between nanoparticles and macrophages, thus more effectively activating macrophages, causing an increase in immune stimulation, which, in turn, improves survival in mice (Figure 4c). This solves the problem of passive therapy requiring multiple injections due to large peritoneal space and rapid excretion. In the direction of immunotherapy, active delivery has broad prospects in the treatment of different types of primary and metastatic peritoneal cavity tumors [50]. Multimodal treatment strategies that are more conducive to the diagnosis and precise treatment of diseases have received widespread attention from researchers. Xing et al. [51] selected marine magnetotactic bacteria (AMB-1) as a template and loaded nano photosensitizers on the bacterial surface by Michael addition reaction to construct an intelligent micro/nano biological robot. Through magnetic/optical sequential manipulation, magnetic navigation, tumor penetration, and photothermal ablation were realized in mice. The results show that micro-nanorobots, under the control of magnetic field, realize the precise migration control of single or group at micron scale and track it in real time through fluorescence and magnetic resonance dual-mode imaging. Using the magnetic and hypoxic integrated target of a micro-na biobot, breaking through the complex physiological barrier with photosensitive agent into the tumor, using remote near-infrared laser trigger to produce local high temperature, the visual precision treatment of tumor is realized (Figure 4d).

The use of micro/nanorobots to transport live cells directly to the target area can improve their retention and survival. Thus, spherical and spiral magnetic micro/nanorobots have been developed for 3D culture and precise delivery of in vitro, discrete, and in vivo stem cells. These miniature robots are manufactured by 3D printing technology and exhibit rolling and spiral motion under the condition of an applied rotating magnetic field, which is more propulsion efficient and more suitable for biofluid than robots driven by magnetic field gradients. Hippocell neural stem cells proliferate on them and differentiate into astrocytes, protrusion glial cells, and neurons. In addition, micro/nanorobots can transfer rectal cancer cells in vitro to tumor microsomic tissue on the microchip of liver tumors. These results show that it is feasible for micro/nanorobots to carry out targeted stem cell transport and transplantation in various in vitro, exosome, and in vivo physiological fluid environments [52]. In addition, a super-magnetic/catalytic micro/nanorobot can move as a single robot and “team up” under the influence of a weak magnetic field to form a chain-shaped spherical structure that can effectively load and transport cancer cells. After loading DOX, they accurately capture breast cancer cells while releasing the drug through diffusion [53]. In addition to stem cell transplantation, micro/nanorobots can also be used as sperm carriers to assist in the fertilization process. For some sperm cells that are low or unable to move due to defective activity, Medina-Sánchez et al. have designed metal-coated-polymer micro-helix robots to transport sperm cells with motion disorders to help them achieve natural fertilization. Fluid channels simulate physiological conditions in which individual inactive living sperm is captured, transported, and released, and individual sperm cells are successfully transported to the cell walls of oocytes. Except for helical microbots, tubular microbots were also designed to capture transport sperm [54]. The advantage of this innovative fertilization method is its potential in vivo applicability, and, if it can target fertilization in the natural environment of oocytes, there is no need to transplant and replace oocytes. However, artificially transporting sperm to oocytes fertilization seems to have a long way to go.

**Figure 4 micromachines-13-00648-f004:**
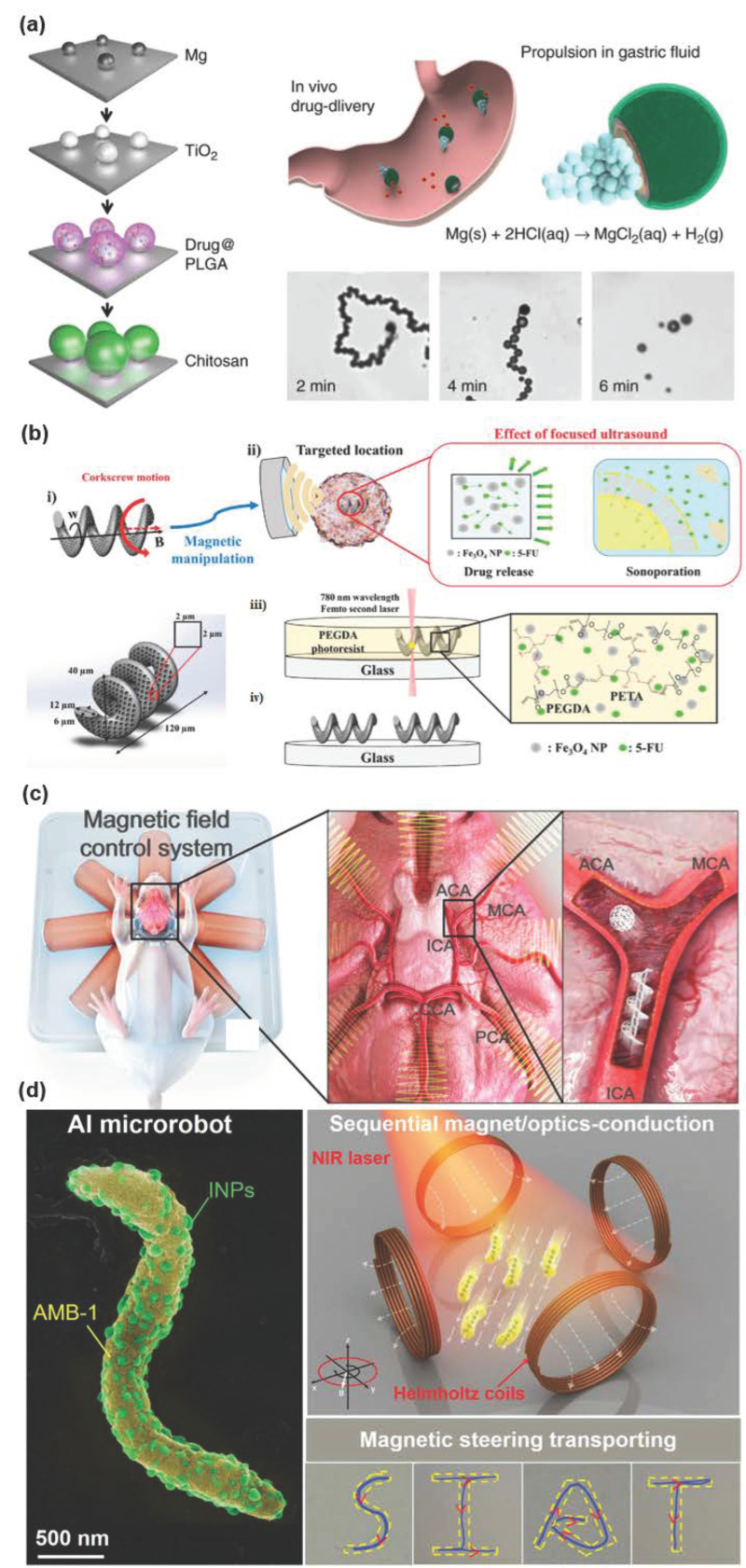
Examples of the application of micro/nanorobots as drug carriers in disease treatment. (**a**) Schematic preparation of the drug-loaded Mg-based micromotors [45], copyright 2017, Springer Nature; (**b**) conceptual schematics of PDM ultrasound-mediated drug treatment [48], copyright 2020, Wiley-VCH; (**c**) magnetic actuation of the microrobots in the brain ex vivo [50], copyright 2019, American Association for the Advancement of Science; (**d**) schematic illustration of biohybrid microrobots. AMB-1 microrobot coupled with INPs, referred to as AI microrobot, with sequential magneto/optics-conducted capability [51], copyright 2020, Wiley-VCH.

### 5.2. Surgery Tool

Traditional surgical tools without micro/nanoscale surgical tools limit the ability to operate on such small scales. Miniaturized micro/nanorobots that can be used as surgical tools will have a clear advantage in reaching areas that are not accessible to catheters and blades. It is also possible to reduce the risk of infection, shorten recovery time, and improve the accuracy and control of surgery.

Using the external physical field and the characteristics of the material itself, micro/nanorobots can directly locate the focus to achieve disease treatment. Chen et al. manipulated magnetic bacteria-microrobots by focusing magnetic fields to target to kill pathogens [55]. The researchers first guided the magnetotactic bacteria-microrobot in the microfluidic chip, and then manipulated the microrobot to target and attach to Staphylococcus aureus. When the microrobot combined with Staphylococcus aureus, the oscillating magnetic field could significantly reduce the viability of Staphylococcus aureus (Figure 5a). Although magnetic targeting device can kill Staphylococcus aureus, simple mixtures or solutions containing only Staphylococcus aureus cannot be killed. These results show that the use of magnetic targeting device is a promising method of targeted therapy of micro/nanorobots. In future research, it is necessary to explore the effects of pulsating blood flow, red blood cells, friction on magnetic bacteria-microrobot control, and the safety of magnetic bacteria in the human body. Resistance to bacteria can also rely on the action of microrobots themselves, such as the use of Ga/Zn micro/nanorobot degradation-produced Ga cations, as built-in antibiotics. Compared with passively used Ga particles, this method enhanced the diffusion of Ga ions and significantly improved the antibacterial efficiency of anti-H. pylori [56]. Cancer cells are more sensitive to heat than normal cells and suffer irreversible thermal damage in environments greater than 40 °C, with temperatures of 42 to 45 °C being enough to kill cancer cells. Therefore, it is possible to use biodegradable thermotherapy micro/nanorobots to convert electromagnetic energy into thermal energy under the action of alternating magnetic fields, as well as to reduce the viability of cancer cells by raising the temperature (Figure 5b). The use of cancer cell lines in vitro has demonstrated the feasibility of this mediated targeted thermal therapy, which kills cancer cells in a way that minimizes damage to the body [47].

The application of injection therapy method in a glass body is expected to promote the development of ophthalmology. The traditional delivery methods rely on the random and passive diffusion of molecules, and they cannot quickly deliver concentrated drugs to the limited area behind the eyes. Most tissues, including vitreous ones, have a close macromolecular matrix as a barrier to prevent the penetration of particles. However, magnetically-driven spiral micro/nanorobots can actively reach the retina through vitreous humor. However, magnetically-driven spiral micro/nanorobots can actively pass-through glass body fluids to reach the retina. Among them, the diameter of the spiral is equivalent to the mesh size of the vitreous biopolymer network, and the surface coating is functionalized with perfluorocarbon. The coating minimizes the interaction between the spiral and the biopolymer (including the collagen bundle in the vitreous) and decreases the adhesion to the surrounding biopolymer network. With wireless excitation from the external magnetic field, large groups of spiral micro/nanorobots can be driven a few centimeters through the eyeball and can reach the retina within 30 min, reducing the delivery time to 1/10 of the original. The entire system is scanned using standard optical coherent faults. Complete operating procedures include glass in vivo injection, remote self-propelled, and noninvasive monitoring [57]. To improve the biocompatibility of micro/nanorobots and minimize the inflammatory response of micro/nanorobots when they enter the eye, Pokki et al. studied polypyrrole (PPy)’s application in gold-coated cobalt-nickel microrobots [58]. They used PPy, which has good long-term biocompatibility with a variety of cells, as a microstructure-protected functional coating and injected a coating-coated microrobot into the rabbit’s eye. The results showed that the biocompatibility of micro/nanorobots was enhanced by the use of PPy coating, and the inflammatory response was minimal compared to the controls not coated with PPy coating. All in all, microbots show potential suitability in drug delivery and retinal prefrontal peeling operations that block retinal veins.

Venous thrombosis, which has a high incidence worldwide, can often be life-threatening. At present, there are obvious shortcomings in the treatment of venous thrombosis, such as short half-life of thrombosis drug treatment, frequent large doses of systemic administration, which brings high medical costs and bleeding, body allergy, and blood pressure instability, and other side effects. Micro/nanobots can load drugs and, with their unique motor ability, release drugs in a controlled manner after deep blood clots, thereby significantly improving the therapeutic effect. Wan et al. have developed a platelet membrane modification, self-moving multi-stage hole nanorobot, used to treat venous thrombosis continuous targeted administration for short-term thrombosis and long-term anticoagulant purposes [59]. At the same time, the micro/nanobots has also made outstanding contributions to the common gastrointestinal problems and gastric wall injury. Bioprinting technology, which delivers new cells directly to the injured site to repair tissue, provides a potentially useful method for the treatment of this problem. The difficulty is that the current bioprinting technology is concentrated outside the human body, and the bioprinter is usually large. Without invasive surgery, it cannot provide enough space for internal tissue repair for printing operation. In order to overcome this problem, Zhao et al. developed a microrobot, which enters the human body through an endoscope for tissue repair in the body. Human gastric epithelial cells and human gastric smooth muscle cells were used as biological ink to simulate the anatomical structure of the stomach. The printed cells still maintained high viability and stable proliferation after 10 days of culture, which showed that the cells had good biological functions in the printed tissue scaffold. This work shows innovative progress not only in the field of biological micro/nanorobots but also in the field of clinical science [60].

**Figure 5 micromachines-13-00648-f005:**
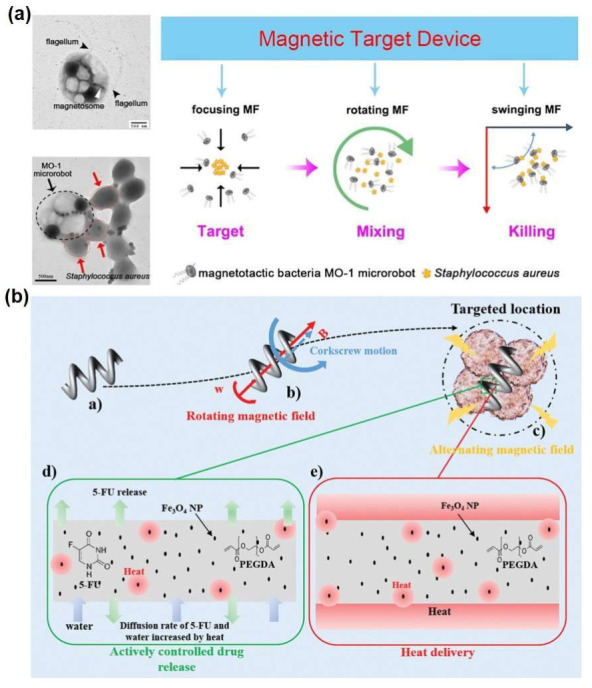
Examples of the application of micro/nanorobots as surgery tools in disease treatment. (**a**) Schematic preparation of the drug-loaded Mg-based micromotors [49], copyright 2019, Elsevier B.V. (**b**) Conceptual diagram of the DHMs [47]. (**a**) A DHM, (**b**) locomotion of the DHM to a target area by an external RMF, (**c**) delivery of a drug and heat to a target area under an AMF, (**d**) actively controlled drug release, and (**e**) delivery of heat energy, copyright 2019, Wiley-VCH.

## 6. Conclusions

Micro/nanorobots have made some breakthroughs in different fields of precision medicine over the past decade due to their tiny size, freedom, and unconstrained and controllable ability to control motion and achieve various functions in any small environment. Micro/nanorobots were used as diagnostic tools to detect ion content in the human body and biosensors; as imaging tools, relying on light, sound, magnetism, and other means of in vivo and in vitro imaging; as a means of transport load and transport drugs, biological reagents and live cells, etc.; and as surgical tools to examine tissues, as well as eliminate cancer cells and bacteria.

Many research studies have been tested on targeted administration in animal trials, but there would still be safety, technical, regulatory and market challenges before micro/nanorobots could be converted into clinical applications. The application of micro/nanorobots has not yet been tested in the human body, and its impact on human health has not yet been assessed. In addition, the preparation and application of micro/nanorobots needs to face the following challenges: (1) Maintaining the function of micro/nanorobots in vivo has been proven to have a strong thrust in the test tube, some of which are still operated in viscous biological fluids (such as gastric juice or whole blood); however, when these microrobots work in physical environments, such as blood vessels, the complex branches of the organism, the dense viscosity of the body fluid, a large number of blood cells, and high fluid speed may bring great challenges to the manipulation of these micro/nanorobots; in addition, the size of the designed micro/nanorobot should be considered when used in vivo. (2) To improve the efficiency of drug delivery, some researchers demonstrated the use of high load capacity carriers to solve this problem; however, the effective load of a single carrier is always limited and cannot fundamentally solve the problem; therefore, a group of micro/nanorobots are expected to reach the target and conduct treatment through independent and coordinated actions with the target of the disease location; the coordinated action of multiple robots can be used to perform tasks that a single robot cannot complete. (3) Removal of the micro/nanorobot from the living body is necessary after completing the task; for practical applications in vivo, degradation and excretion of drug delivery systems are also very important; micro/nanorobots must be degraded into non-toxic compounds easily without external interference through passive or built-in self-destructive mechanisms after completing the task. (4) To manufacture intelligent micro/nanorobots, an ideal drug delivery system should combine specific sensing and drug delivery functions, such as some active microorganisms, and be used for diagnosis and treatment; intelligent and multifunctional micro/nanorobots are expected to open a new era in which the targeted drug delivery method based on micro/nanorobots will become an important way of active drug delivery.

Although large-scale clinical applications of micro/nanorobots will encounter certain difficulties, the potential for the use of microbots in precision medicine to diagnose and treat diseases is enormous. Once micro/nanorobots are initially validated among human subjects to help with precision medicine, reduce costs, and reduce the pain of surgery, it will greatly advance the development of modern biomedical science and, to a large extent, change human life.

## Figures and Tables

**Figure 1 micromachines-13-00648-f001:**
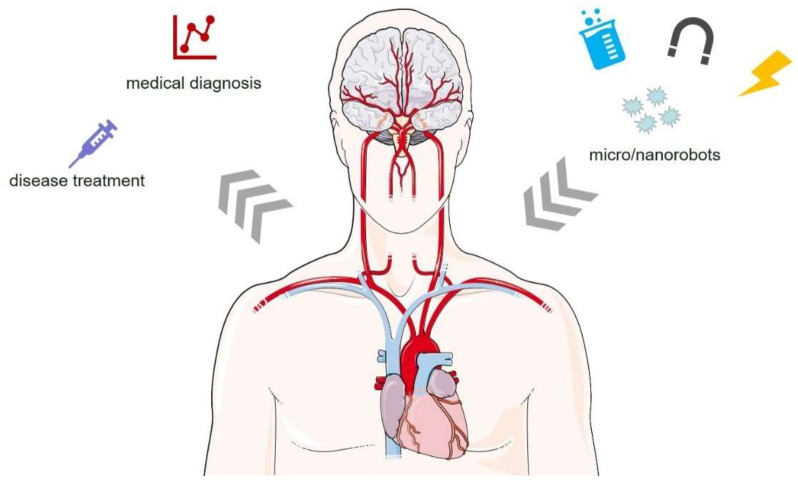
Schematic view of micro/nanorobots for medical diagnosis and disease treatment.

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
