# Peer review of "Micro/Nanorobots for Medical Diagnosis and Disease Treatment"

_micromachines, 2022, doi:10.3390/mi13050648_

Round 1
Reviewer 1 Report
The author reviewed the driving methods of micro/nano robots and their medical applications, and discussed the challenges and future development prospects of their clinical applications. This review has certain reference value for the researchers preparing to carry out the research of micro/nano robots for medical diagnosis.
There are some minor problems in the article, and further modifications are suggested.
1. As an review, the author needs to help the reader grasp the main points of the references, so it is necessary to present an overview at the beginning of each chapter. I would suggest that there should be such a paragraph at the begining of the Chapter 3 and 4.
2. The author should emphasize the comparison and commentary of the references in the last paragraph of each chapter and the conclusion. The summary is somewhat empty and lacks substantive conclusions.
3. There is a typo on page 14. The chapter number of the conclusion should be 5.
Author Response
Response to Reviewer 1 Comments
Point 1: As an review, the author needs to help the reader grasp the main points of the references, so it is necessary to present an overview at the beginning of each chapter. I would suggest that there should be such a paragraph at the begining of the Chapter 3 and 4.
Response 1: Thanks for your good suggestions. The overview at the begining of the Chapter 3 and 4 have been added.
Point 2: The author should emphasize the comparison and commentary of the references in the last paragraph of each chapter and the conclusion. The summary is somewhat empty and lacks substantive conclusions.
Response 2: Thanks for your good suggestions. Relevant content has been enriched.
Point 3: There is a typo on page 14. The chapter number of the conclusion should be 5.
Response 3: Thanks for your good suggestions. Relevant content has been enriched.

Reviewer 2 Report
The review paper introduce relevant driving methods of micro/nanorobots preparation in details, summarizes the progress of research in medical applications, and discusses the challenges it faces in clinical applications and the future direction of development. I think this paper is valuable to the researchers in the relevant areas. I suggest publication of this paper.
Author Response
Response to Reviewer 2 Comments
Point 1: The review paper introduce relevant driving methods of micro/nanorobots preparation in details, summarizes the progress of research in medical applications, and discusses the challenges it faces in clinical applications and the future direction of development. I think this paper is valuable to the researchers in the relevant areas. I suggest publication of this paper.
Response 1: Thank you for your consideration.

Reviewer 3 Report
This manuscript reports a review about micro/nanorobots for medical diagnosis and disease treatment. However, this manuscript can be significantly improved considering the following comments:
1.-English grammar and style of all the sections of manuscript must be improved.
2.-This review is very short and has few sections, references and figures about topics of micro/nanorobots for potential applications in biomedicine. This manuscript should add more sections and discussion about operation principle and design, fabrication process, signal processing, and challenges of micro/nanorobots.
3.-Abstract must include more information about contribution of the different sections of the manuscript and the main advantages and potential applications of micro/nanorobots in biomedical field. In addition, the following sentences must be modified:
Micro/nanorobots are functional devices manufactured with microns, nanoscale,
The term manufactured should be changed. For instance, the following sentence could be used: Micro/nanorobots are functional devices in micro and nanoscale,...
Also, the following sentence is ambiguous:
For potential to achieve more efficient and accurate local diagnosis and treatment, there are broad application prospects in biomedical fields.
4.-Introduction has various mistakes. For instance,
Richar biofluid Feynman
micron nanomaterials
In addition, introduction should add more information of the advantages, limitations and potential applications of micro/nanorobots in biomedicine. Also, this section could include more references and information about micro/nanorobots in biomedicine.
Scheme should be Figure 1. For instance, Figure 1. Schematic view of Micro/nanorobots for medical diagnosis and disease treatment.
Also, this figure should change the image of conventional robot by a better image about micro/nanorobot.
5.-Various sentences have many words, which complicate their reading. The redaction of the following sentences must be improved.
Line 76-79
Therefore, it is necessary to implement the actual application, especially in the biological system to drive the micro/ nano robot, and need to determine the new in situ fuel than H2O2, that is, the material should be the natural substance in the biological fluid, not added from the outside.
Line 85
micro/nanorobot robots
Line 87-88
They are more flexible in terms of control movement, and the acoustic conditions are easier.
Line 97
amphoto-driven isotropic TiO2
Line 170-173
The micro/nanorobot accelerated the spread of solute and fully mixed with the target, and its magnetic meconium silica microtube provided a rich load space for the adsorption of the active substance tetramethylene pentaamine (TEPA), thus showing good adsorption ability and short processing time to Cu2+.
Line 196-201
The study showed that with millimoles in human serum, the rapid motion of the Mg/Pt Janus micro/nanorobot enhanced the current signal and the current signal increased with the number of micro/nanorobot introduced, the heartbeat improved, and the addition of the micro/nanorobot established a linear relationship between the current signal and the glucose concentration.
Line 237-241
Among them, the micro/nanorobots have a functional multi-layer coating, its Au layer is used to enhance light absorption and increase the rate of propulsion, gelatin hydrogel layer is used to expand the load capacity of different functional components, polyethylene layer is used to maintain the geometry of the micro/nanorobots during propulsion.
Line 256-261
Because microalgae have inherent properties that allow fluorescence imaging in vivo, the spiral robot does not require any surface modification to show intrinsic fluorescence, magnetic resonance signals, natural degradation, and ideal cytotoxicity, is capable of robust navigation in a variety of biological fluids, and can be tracked noninvasively through autofluorescence, magnetic resonance, as in shallow tissues or deep organs.
Line 270-273
A drug-carrying micro/nanorobot for the treatment of stomach bacterial infections, shown in Figure 3a, has an average size of about 20 .m of Mg particles at its core and can advance efficiently, with an average speed of about 120 .m/s tested in vitro simulated gastric fluid
Line 273-278
The in vitro sterilization activity test of Helicobacter pylori showed that, within the whole concentration range used in the study, the drug-carrying robot showed the same sterilization activity as the free drug solution, and the micro/nanorobot could be effectively promoted and distributed throughout the stomach of live mice, significantly reducing the number of Helicobacter pylori.
Line 352-357
Xing et al. used ocean-sourced magnetomagnetic bacteria (AMB-1) as a template, using Michael's additive reaction to load nano-photosensitive agents onto bacterial surfaces, and built intelligent microna biobots that, through magnetic/optical sequential manipulation, achieved magnetically controlled navigation, tumor penetration and photothermal ablation inmice [43].
Line 408-411
The researchers first guided the magnetizing bacteria-micro-robot in a microfluidic chip, and then manipulated the microbot to target the acid, which, when combined with Staphylococcus aureus, applied a swinging magnetic field that significantly reduced Staphylococcus aureus viability (Figure 4a).
Line 430-433
Traditional delivery methods rely on random, passive diffusion of molecules, cannot quickly deliver concentrated drugs to defined areas behind the eyes, and most tissues, including glass bodies, have a tight macromolecular matrix as a barrier to prevent particles from penetrating.
Line 435-439
Among them, the diameter of the spiral is comparable to the mesh size of the biopolymer network of the glass body, and the surface coating is functionalized with perfluorocarbons, which minimizes the interaction of spirals and biopolymers (including collagen bundles in vitreous) to minimize adhesion to the surrounding biopolymer network.
Line 455-458
Current treatment of venous thrombosis has some obvious disadvantages, including the short half-life of the treatment of thrombotic drugs, so frequent large doses of systemic administration, which brings high medical costs and side effects (bleeding, body allergies and blood pressure instability, etc.) harm.
7.-Line 193, authors must indicate the reference of Kong et al.
For instance, Kong et al. [reference]
Line 203, authors must indicate the reference of Li et al.
Line 244, authors must indicate the reference of Iacovacci et al.
Line 319, authors must indicate the reference of Zhong et al.
8.-Conclusion must be written in past tense.
9.-Resolution of Figures 1-4 must be improved.
10.-This review has few figures and references about micro/nanorobots. Authors must add more figures and references about the operation principles, design, fabrication, and potential applications of micro/nanorobots in biomedicine.
Author Response
Response to Reviewer 3 Comments
This manuscript reports a review about micro/nanorobots for medical diagnosis and disease treatment. However, this manuscript can be significantly improved considering the following comments:
Point 1: English grammar and style of all the sections of manuscript must be improved.
Response 1: Thanks for your good suggestions. English grammar and mistakes have been revised.
Point 2: This review is very short and has few sections, references and figures about topics of micro/nanorobots for potential applications in biomedicine. This manuscript should add more sections and discussion about operation principle and design, fabrication process, signal processing, and challenges of micro/nanorobots.
Response 2: Thanks for your good suggestions. The discussion on the preparation methods of micro/nanorobots and the challenges in the future has been added.
Point 3: Abstract must include more information about contribution of the different sections of the manuscript and the main advantages and potential applications of micro/nanorobots in biomedical field. In addition, the following sentences must be modified:
Micro/nanorobots are functional devices manufactured with microns, nanoscale,
The term manufactured should be changed. For instance, the following sentence could be used: Micro/nanorobots are functional devices in micro and nanoscale,...
Also, the following sentence is ambiguous:
For potential to achieve more efficient and accurate local diagnosis and treatment, there are broad application prospects in biomedical fields.
Response 3: Thanks for your good suggestions. Abstract has been enriched. “Micro/nanorobots are functional devices manufactured with microns, nanoscale” has revised as “Micro/nanorobots are functional devices in micro and nanoscale”.And the sentence” For potential to achieve more efficient and accurate local diagnosis and treatment, there are broad application prospects in biomedical fields.”has been corrected to “Micro/nanorobots are expected to achieve more efficient and accurate local diagnosis and treatment, and have broad application prospects in the biomedical field.”
Point 4: Introduction has various mistakes. For instance,
Richar biofluid Feynman
Response 4-1: Thanks for your good suggestions. It has been revised as Richard Phillips Feynman.
micron nanomaterials
Response 4-2: Thanks for your good suggestions. It has been revised as micro/nanomaterials
In addition, introduction should add more information of the advantages, limitations and potential applications of micro/nanorobots in biomedicine. Also, this section could include more references and information about micro/nanorobots in biomedicine.
Response 4-3: Thanks for your good suggestions. The introduction has been supplemented.
Scheme should be Figure 1. For instance, Figure 1. Schematic view of Micro/nanorobots for medical diagnosis and disease treatment.
Response 4-4: Thanks for your good suggestions. It has been revised.
Also, this figure should change the image of conventional robot by a better image about micro/nanorobot.
Response 4-5: Thanks for your good advice. Figure 1 has been revised.
Point 5: Various sentences have many words, which complicate their reading. The redaction of the following sentences must be improved.
Line 76-79
Therefore, it is necessary to implement the actual application, especially in the biological system to drive the micro/nanorobots, and need to determine the new in situ fuel than H2O2, that is, the material should be the natural substance in the biological fluid, not added from the outside.
Response 5-1: Thanks for your good suggestions. It has been revised as “Therefore, in order to realize practical applications, especially when chemical driving is used to drive micro/nanorobots in biological systems, it is necessary to develop new in-situ fuels other than H2O2, that is, the raw materials should be natural substances in biological fluids”.
Line 85
micro/nanorobot robots
Response 5-2: Thanks for your good suggestions. It has been revised as “micro/nanorobots”.
Line 87-88
They are more flexible in terms of control movement, and the acoustic conditions are easier.
Response 5-3: Thanks for your good suggestions. It has been revised as “Most of the external field driven micro/nanorobot robots do not require fuel and are driven primarily by light, ultrasonic or magnetic fields, making them both bio-compatibility and sustainable, and they are more flexible in terms of control movement. It is relatively easy to establish acoustic conditions”.
Line 97
amphoto-driven isotropic TiO2
Response 5-4: Thanks for your good suggestions. It has been revised as “photo-driven isotropic TiO2”.
Line 170-173
The micro/nanorobot accelerated the spread of solute and fully mixed with the target, and its magnetic meconium silica microtube provided a rich load space for the adsorption of the active substance tetramethylene pentaamine (TEPA), thus showing good adsorption ability and short processing time to Cu2+.
Response 5-5: Thanks for your good suggestions. It has been revised as “The micro/nanorobot accelerated the spread of solute and fully mixed with the target, and its magnetic meconium silica microtube provided a rich load space for the adsorption of tetramethylene pentaamine (TEPA), thus showing good adsorption ability and short processing time for Cu2+”.
Line 196-201
The study showed that with millimoles in human serum, the rapid motion of the Mg/Pt Janus micro/nanorobot enhanced the current signal and the current signal increased with the number of micro/nanorobot introduced, the heartbeat improved, and the addition of the micro/nanorobot established a linear relationship between the current signal and the glucose concentration.
Response 5-6: Thanks for your good suggestions. It has been revised as “The study showed that the rapid movement of the Mg/Pt Janus micro/nanorobot enhanced the current signal and the current signal increased with the increase of introduced micro/nanorobots, and the addition of micro/nanorobots established great linear relationship between the current signal and the glucose concentration”.
Line 237-241
Among them, the micro/nanorobots have a functional multi-layer coating, its Au layer is used to enhance light absorption and increase the rate of propulsion, gelatin hydrogel layer is used to expand the load capacity of different functional components, polyethylene layer is used to maintain the geometry of the micro/nanorobots during propulsion.
Response 5-7: Thanks for your good suggestions. It has been revised as “Among them, the micro/nanorobot has multi-layer functional coating, its Au layer enhanced the optical absorption and improved the propulsion rate, gelatin hydrogel layer expanded the load capacity of different functional components, and polyethylene layer maintained the robot's geometry in the process of propulsion”.
Line 256-261
Because microalgae have inherent properties that allow fluorescence imaging in vivo, the spiral robot does not require any surface modification to show intrinsic fluorescence, magnetic resonance signals, natural degradation, and ideal cytotoxicity, is capable of robust navigation in a variety of biological fluids, and can be tracked noninvasively through autofluorescence, magnetic resonance, as in shallow tissues or deep organs.
Response 5-8: Thanks for your good suggestions. It has been revised as “Since microalgae have the advantages of in vivo fluorescence imaging, the spiral robot showed intrinsic fluorescence, magnetic resonance signals, natural degradation and ideal cytotoxicity without any surface modification. It can stably navigate in various biological fluids, and track non-invasively in superficial tissues or deep organs through autologous fluorescence and magnetic resonance imaging”.
Line 270-273
A drug-carrying micro/nanorobot for the treatment of stomach bacterial infections, shown in Figure 3a, has an average size of about 20 .m of Mg particles at its core and can advance efficiently, with an average speed of about 120 .m/s tested in vitro simulated gastric fluid
Response 5-9: Thanks for your good suggestions. It has been revised as “A drug-loaded micro/nanorobot for the treatment of gastric bacterial infection is made of Mg particles with an average size of about 20 μm shown in Figure 3a. It has the ability of efficient propulsion, and the average velocity tested in vitro simulated gastric juice (pH = 1.3) is about 120 μm/s”.
Line 273-278
The in vitro sterilization activity test of Helicobacter pylori showed that, within the whole concentration range used in the study, the drug-carrying robot showed the same sterilization activity as the free drug solution, and the micro/nanorobot could be effectively promoted and distributed throughout the stomach of live mice, significantly reducing the number of Helicobacter pylori.
Response 5-10: Thanks for your good suggestions. It has been revised as “In vitro bactericidal activity test of Helicobacter pylori (H. pylori) showed that the drug-loaded robot showed similar bactericidal activity to free drug solution, and the micro/nanorobot could be effectively promoted and distributed throughout the stomach of live mice, thereby significantly reducing the number of H. pylori”.
Line 352-357
Xing et al. used ocean-sourced magnetomagnetic bacteria (AMB-1) as a template, using Michael's additive reaction to load nano-photosensitive agents onto bacterial surfaces, and built intelligent microna biobots that, through magnetic/optical sequential manipulation, achieved magnetically controlled navigation, tumor penetration and photothermal ablation inmice [43].
Response 5-11: Thanks for your good suggestions. It has been revised as “Xing et al. selected marine magnetotactic bacteria (AMB-1) as a template and loaded nano photosensitizers on the bacterial surface by Michael addition reaction to construct an intelligent micro-nano biological robot. Through magnetic / optical sequential manipulation, magnetic navigation, tumor penetration and photothermal ablation were realized in mice”.
Line 408-411
The researchers first guided the magnetizing bacteria-micro-robot in a microfluidic chip, and then manipulated the microbot to target the acid, which, when combined with Staphylococcus aureus, applied a swinging magnetic field that significantly reduced Staphylococcus aureus viability (Figure 4a).
Response 5-12: Thanks for your good suggestions. It has been revised as “The researchers first guided the magnetotactic bacteria-microrobot in the microfluidic chip, and then manipulated the microrobot to target and attach to Staphylococcus aureus. When the microrobot combined with Staphylococcus aureus, the oscillating magnetic field could significantly reduce the viability of Staphylococcus aureus”.
Line 430-433
Traditional delivery methods rely on random, passive diffusion of molecules, cannot quickly deliver concentrated drugs to defined areas behind the eyes, and most tissues, including glass bodies, have a tight macromolecular matrix as a barrier to prevent particles from penetrating.
Response 5-13: Thanks for your good suggestions. It has been revised as “The traditional delivery methods rely on the random and passive diffusion of molecules, and cannot quickly deliver concentrated drugs to the limited area behind the eyes. Most tissues, including vitreous, have a close macromolecular matrix as a barrier to prevent the penetration of particles. However, magnetically driven spiral micro / nanorobots can actively reach the retina through vitreous humor”.
Line 435-439
Among them, the diameter of the spiral is comparable to the mesh size of the biopolymer network of the glass body, and the surface coating is functionalized with perfluorocarbons, which minimizes the interaction of spirals and biopolymers (including collagen bundles in vitreous) to minimize adhesion to the surrounding biopolymer network.
Response 5-14: Thanks for your good suggestions. It has been revised as “Among them, the diameter of the spiral is equivalent to the mesh size of the vitreous biopolymer network, and the surface coating is functionalized with perfluorocarbon. The coating minimizes the interaction between the spiral and the biopolymer (including the collagen bundle in the vitreous), and decreases the adhesion to the surrounding biopolymer network”.
Line 455-458
Current treatment of venous thrombosis has some obvious disadvantages, including the short half-life of the treatment of thrombotic drugs, so frequent large doses of systemic administration, which brings high medical costs and side effects (bleeding, body allergies and blood pressure instability, etc.) harm.
Response 5-15: Thanks for your good suggestions. It has been revised as “At present, there are obvious shortcomings in the treatment of venous thrombosis, such as short half-life of thrombosis drug treatment, frequent large doses of systemic administration, which brings high medical costs and bleeding, body allergy and blood pressure instability and other side effects”.
Point 7: Line 193, authors must indicate the reference of Kong et al.
For instance, Kong et al. [reference]
Line 203, authors must indicate the reference of Li et al.
Line 244, authors must indicate the reference of Iacovacci et al.
Line 319, authors must indicate the reference of Zhong et al.
Response 7: Thanks for your good suggestions. The references have been indicated.
Point 8: Conclusion must be written in past tense.
Response 8: Thanks for your good suggestions. The tense of Conclusion has been corrected.
Point 9: Resolution of Figures 1-4 must be improved.
Response 9: Thanks for your good suggestions. The resolution of Figures has been improved.
Point 10: This review has few figures and references about micro/nanorobots. Authors must add more figures and references about the operation principles, design, fabrication, and potential applications of micro/nanorobots in biomedicine.
Response 10: Thanks for your good suggestions. The discussion on the preparation methods of micro/nanorobots and the challenges in the future has been added.

Round 2
Reviewer 3 Report
This revised manuscript was improved considering the comments of reviewer.
This manuscript is a resubmission of an earlier submission. The following is a list of the peer review reports and author responses from that submission.